# Investigation on Fermentation Characteristics and Microbial Communities of Wheat Straw Silage with Different Proportion *Artemisia argyi*

**DOI:** 10.3390/toxins15050330

**Published:** 2023-05-11

**Authors:** Zhenyu Wang, Zhongfang Tan, Guofang Wu, Lei Wang, Guangyong Qin, Yanping Wang, Huili Pang

**Affiliations:** 1Henan Key Laboratory Ion Beam Bioengineering, School of Agricultural Sciences, Zhengzhou University, Zhengzhou 450052, China; 202022582017499@gs.zzu.edu.cn (Z.W.); tzhongfang@zzu.edu.cn (Z.T.); qinguangyong@zzu.edu.cn (G.Q.); wyp@zzu.edu.cn (Y.W.); 2Plateau Livestock Genetic Resources Protection and Innovative Utilization Key Laboratory of Qinghai Province, Key Laboratory of Animal Genetics and Breeding on Tibetan Plateau, Ministry of Agriculture and Rural Affairs, Qinghai Academy of Animal and Veterinary Medicine, Qinghai University, Xining 810016, China; jim963252@163.com (G.W.); wanglei382369@163.com (L.W.)

**Keywords:** *Artemisia argyi*, wheat straw, mycotoxins, fermentation characteristic, microbial community

## Abstract

Mycotoxins, secondary metabolites of fungi, are a major obstacle to the utilization of animal feed for various reasons. Wheat straw (WS) is hollow, and miscellaneous bacteria can easy attach to its surface; the secondary fermentation frequency after silage is high, and there is a risk of mycotoxin poisoning. In this study, a storage fermentation process was used to preserve and enhance fermentation quality in WS through the addition of Artemisia argyi (AA), which is an effective method to use WS resources and enhance aerobic stability. The storage fermentation of WS treated with AA had lower pH and mycotoxin (AFB1 and DON) values than the control due to rapid changes in microbial counts, especially in the 60% AA groups. Meanwhile, the addition of 60% AA improved anaerobic fermentation profiles, showing higher lactic acid contents, leading to increased efficiency of lactic acid fermentation. A background microbial dynamic study indicated that the addition of 60% AA improved the fermentation and aerobic exposure processes, decreased microbial richness, enriched *Lactobacillus* abundance, and reduced *Enterobacter* and *Aspergillus* abundances. In conclusion, 60% AA treatment could improve the quality by increase fermentation quality and improve the aerobic stability of WS silage by enhancing the dominance of desirable *Lactobacillus*, inhibiting the growth of undesirable microorganisms, especially fungi, and reducing the content of mycotoxins.

## 1. Introduction

As secondary metabolites of fungi, mycotoxins are hazardous to both humans and livestock when fungi attack crops and infect their products. Frighteningly, mycotoxins pollute 60–80% of crops [1]. Cereal crops, including wheat, one of the most important crops for human nutritional security, livestock feed, and international trade, are extremely vulnerable to fungal infection [2]. In particular, rapid global climate change and shifts in cultivation systems, such as the promotion of direct return of straw to the field rather than burning, have led to the frequent occurrence of fungal diseases such as *Fusarium* head blight (FHB) caused by *Fusarium* spp. On the basis of the classification by the International Agency for Research on Cancer (IARC), *Fusarium* spp. can produce a variety of different mycotoxins such as deoxynivalenol (DON) and zearalenone (ZEN), among which DON is the most common [3]. Aflatoxin B1(AFB1), one of the secondary metabolites of *Aspergillus*, belonging to Group 1 according to the IARC, has carcinogenicity to humans and is commonly found in moldy grains, straw, and feed. AFB1 can be discharged from livestock byproducts and accumulated in the food chain constantly, leading to carcinogenic, mutagenic, and teratogenic phenomena, causing great harm [4]. The existence of mycotoxins has not only reduced the yield of wheat in a large area, seriously affecting its quality and causing major economic losses, but also brought great challenges to the feed utilization of wheat straw (WS) [5].

WS, the main by-product of wheat planting, has become an inevitable choice for developing roughage resources because of its abundant resources and low price [6]. Biological treatment can successfully solve problems including the low content of available nutrients and poor palatability of WS, among which ensiling is the most effective technology used to preserve the nutritional value of dry straw [7]. Scientific and reasonable collection and ensiling processes can effectively reduce the content of mycotoxins [8,9,10,11]. In addition to selecting WS varieties with less disease, such as FHB, it is necessary to strengthen the management of silage processes to reduce the toxin pollution of feed. Reducing the occurrence of secondary fermentation in the open-feeding process after the completion of silage is also an important means to inhibit the growth of fungi and reduce the production of mycotoxins [9,10].

Physical, chemical, and biological methods have been used to solve issues related to aerobic stability and fermentation quality [12,13]. *Lentilactobacillus buchneri* (LB), mentioned by Muck [14], is a potential additive that may help to enhance aerobic stability. It has evolved into the predominant species employed in obligatory heterofermentative lactic acid bacteria (LAB) inoculants [15]. Meanwhile, cellulase degrades the cell walls (or cellulose) of the material and releases available substrates for fermentation with LAB, which is used to improve the silage stability of high-fiber content silage materials [16]. Nevertheless, the activity and proportion of LAB and cellulase preparation are often unpredictable, and these methods have not progressed sufficiently to improve WS quality. Thus, it is vital to develop new silage additives that can effectively improve quality and control aerobic stability to assure silage quality.

In eastern Asia, a large genus of herbal plants called *Artemisia argyi* (AA) is extensively distributed in wild areas and grasslands, where it is frequently utilized as a functional food and herbal remedy for a variety of ailments [17]. Extracts and active compounds of AA show excellent antioxidant activity and bacteriostasis, which can improve the antioxidant ability of silage, decrease the pH value, ammonia nitrogen (NH_3_-N) content, and number of harmful microorganisms, and significantly lower the level of mycotoxins including AFB1, DON, ZEN, ochratoxin (OA), and fumonisin (FUM) [18]. Recently, AA has been increasingly used as a feed additive or fermentation raw material in animal husbandry. Wang et al. [19] added AA to whole corn straw silage, revealing that the addition of AA could reduce the levels of DON and FUM in the feed, as well as significantly improve the silage quality during aerobic exposure. Kim et al. [20] proposed that AA could regulate gastrointestinal microorganisms and improve the growth performance and stability of meat; replacing straw with AA in sheep diet significantly improved feed intake, rumen fermentation, internal digestibility, nitrogen retention, and microbial nitrogen yield, especially at medium and high AA content [21]. Moreover, several studies have shown that AA or its extracts can be employed as a feedstuff or supplement to boost body metabolism and improve the production performance of livestock [22,23].

To sum up, mixing AA and WS together for silage may enhance the fermentation quality and reduce the mycotoxin production of WS silage. Consequently, this study aimed to explore the effects of different proportions of AA on microbial community, fermentation characteristics, and mycotoxins of WS through ensiling, as well as during the post-opening exposure process, leading to the selection of the optimal addition of AA, obtention of high-quality WS silage, and provision of a reference for solving the mycotoxin problem of fermented feed.

## 2. Results and Discussion

### 2.1. Characteristics of Fresh Materials before Storage Fermentation

Table 1 shows the fermentation quality, chemical composition, and microbial population of WS and AA. The DM content of raw WS was 94.07%, while that of AA was 34.19%, satisfying the ideal content of 30–35% for good silage [24]. Generally, 60–70 g/kg DM WSC is necessary for well-preserved fermentation, being able to provide sufficient substrate for lactic acid fermentation [25,26]. The WSC in fresh AA was 62.13 g/kg DM, sufficient to ensure the success of the silage process and improve the fermentation quality. Compared with WS, AA had relatively higher contents of WSC and CP, but lower contents of NDF and ADF, indicating that the mixed silage could provide enough nutrients.

The initial LAB population in the raw material was 5.00 log cfu/g FM, representing the minimum value to guarantee the acquisition of good-quality silage [14]. A higher number LAB in AA can effectively decrease the pH value in the fermentation process, inhibit the growth and reproduction of undesirable microorganisms, and lower the content of mycotoxins, including AFB1 and DON. However, the initial populations of Bacillus and aerobic bacteria were both above 7.10 log cfu/g FM, and the AFB1 levels in WS and AA were 2.96 µg/kg and 4.14 μg/kg, while the DON levels were 0.08 mg/kg and 0.11 mg/kg, respectively. These might all be barriers to direct fermentation without exogenous additives or feeding without treatment [27].

### 2.2. Characteristics of Different Silage Treatments during Storage Fermentation and Aerobic Exposure

#### 2.2.1. Fermentation Quality

Through 120 days of anaerobic fermentation, the pH in all groups remained below 5.00 (Table 2). After the silage was opened and exposed to the air, heat was generated due to the decomposition of carbohydrates, proteins, and amino acids by aerobic microorganisms [28], while the pH increased, exceeding 5.00 in WS and CE silages after 2 days of aerobic exposure, while remaining below 4.00 in AA. This shows that the addition of AA delayed the rise in pH value. With a prolonged aerobic exposure time, pH values significantly increased in all groups (*p* < 0.05); up to 8 days of aerobic exposure, the pH of the LB, CL, 40% AA, and 60% AA groups was always lower than 5.00, being lowest in the 60% AA group, where it was 4.85 at the end of fermentation after 12 days of exposure. A silage of good fermentation quality can only proceed smoothly if the pH, the most fundamental parameter for evaluating the quality of silage fermentation, is maintained in an optimal range [19]. Values lower than 5.00 are appropriate for crop feedstocks with sugar contents throughout the fermentation process [29]. The WS storage fermentation was facilitated by mixing with 60% AA. This may have contributed to 60% AA exerting an antibacterial impact, reducing the growth of aerobic bacteria and, thus, preventing the rapid increase in pH [30].

The NH_3_-N content is correlated with the degree of protein deterioration [31]. A higher content of NH_3_-N shows that there is a significant amount of protein degradation, probably due to fermentation by *Clostridium* or *Enterobacter* [32]. In this research, at the end of fermentation, the content of NH_3_-N increased with the increase in AA applied content, probably due to the higher CP content of AA itself. The NH_3_-N content in AA-treated groups decreased, especially in 60% AA, until the end of aerobic exposure. This result was in line with the decrease in undesirable microorganisms such as mold and coliform bacteria. The lower NH_3_-N concentration in silage with 60% AA contributed to efficient nitrogen utilization. This phenomenon cab be attributed to low *Clostridium* fermentation as a result of the bacteriostatic and bactericidal action of organic acids and added AA [33].

Lactic acid, a main product of LAB, causes a drop in pH early in the fermentation process [34]. In this study, although pH values decreased following the addition of AA and the prolongation of fermentation days, the lactic acid content changed only slightly, possibly due to the low content of WSC in WS. After aerobic exposure for 4 days, the maximum lactic acid value of ~5.79 g/kg DM was obtained in the 60% AA group. The fermentation quality of 60% AA could contribute to a higher number of LAB, which increased the lactic acid concentration, reduced the pH value, and improved the fermentation quality throughout the process, especially at the aerobic exposure stage [35]. In contrast, the lactic acid and acetic acid contents in WS and CE were lower during aerobic exposure, probably because unwanted microorganisms that were not effectively suppressed consumed nutrients, thus reducing acid content.

#### 2.2.2. Chemical Composition

There was no significant variation in the DM content of fermented silages between the control and treatment groups (Table 3). The WSC content is critical for lactic acid fermentation [36]. At the end of fermentation in this study, the WSC content in all groups decreased, and the lowest content was found in LB. During aerobic exposure, the slowest decrease in WSC content was observed in 60% AA silage, through inhibition of the undesirable bacteria to reduce WSC consumption [37]. With the extension of ensiling, NDF and ADF contents decreased in CE group, while, in CL group, NDF was significantly reduced. More significantly, the NDF and ADF contents in the 60% AA groups were relatively lower than the CE and CL contents at the end of aerobic exposure. The decline in NDF, CE, and CL observed in the 60% AA group suggested an improvement in the palatability and digestibility of 60% AA-treated WS silage [38].

In the process of fermentation, microorganisms consume part of the organic matter through respiration, releasing CO_2_ and H_2_O, which manifests as a reduction in CP content [39]. CP degradation in the silage caused by infectious microbes, such as *Clostridium* and *Enterobacter*, is more likely to occur in the absence of inoculation with foreign microbes [40]. At the end of aerobic exposure, CP loss in the 60% AA group remained the lowest, thereby inhibiting the proliferation of harmful bacteria and reducing their consumption of nutrients such as CP, due to the lower pH [41,42].

#### 2.2.3. Microbial Population

The microbial population data are shown in Table 4. The number of LAB in the 60% AA group was the highest with 7.27 log cfu/g of FM when determined after 120 days, and this was maintained over time with aerobic exposure; meanwhile, the levels of both coliform bacteria and Bacillus were the lowest. Fast acidification might be the key to accelerating the growth of LAB and controlling the growth of harmful bacteria [43,44]. In this study, an increase in the LAB produced copious lactic acid and a rapid decrease in the pH value, in addition to enhancing the fermentation quality of 60% AA. With prolonged exposure, LB, CL, and all AA groups had an obvious inhibitory effect on fungi (*p* < 0.05). Generally, AA inhibits the growth of hazardous microorganisms including *Escherichia*, *Colletotrichum*, and *Aspergillus* [45,46], consistent with the results in this study, where the amounts of Bacillus and aerobic bacteria in AA-treated groups were lower at the end of aerobic exposure, especially in the 60% AA group.

Yeast propagates under aerobic conditions; it can decompose various organic acids and lactic acid to compete for WSC content with LAB, damaging the silage environment [14,47]. No yeast was detected in the AA-treated groups after 120 days of fermentation in this study. Furthermore, after 2 days of aerobic exposure, only the 60% AA group had no yeast. With the extension of aerobic exposure time, the yeast count in the 60% AA group remained at a minimal level until the end of the experiment. The yeast inhibition could also be attributed to the higher lactic acid content in 60% AA.

### 2.3. Mycotoxin Contents of Different Treatments during Aerobic Exposure

Combining the above results, the 60% AA- and CL-treated groups were selected for an analysis of the mycotoxin contents and microbial communities. Mycotoxins in feed are associated with lower fertility, increased disease susceptibility, and animal health problems [48]. Mycotoxins are secondary metabolites of fungi that remain in fermented feed long after the fungi have died, continuing to pose a health hazard to both animals and humans. The transformation of mycotoxins into nontoxic metabolites by using microorganisms and Chinese herbal medicine based on biodegradation enzymes or by directly adding bioenzymes is currently attracting widespread interest from researchers due to the advantages of high efficiency, specificity, and environmental friendliness [49]. The results of this study showed that the AFB1 and DON contents increased in all groups; they were the lowest after 120 days of ensiling in the 60% AA group (Figure 1). The increase in mycotoxin contents after fermentation could have been due to mold activity and mycotoxin production in the early stage. With the extension of silage, the pH decreased further, and LAB took over as the dominant flora; accordingly, mold was limited, and mycotoxin production gradually decreased, but still accumulated [8]. AFB1 and DON are closely related to the structure of the fungal community [50]; thus, the higher amount of LAB and the lower number of unwanted microorganisms in the 60% AA group may have contributed to the lower mycotoxin content.

With the extension of aerobic exposure, the mycotoxin content increased in all groups, especially in WS, while the content in the 60% AA group was still the lowest. Moreover, over 12 days of aerobic exposure, the contents of AFB1 and DON in all groups significantly increased. They were highest for WS, but remained lowest in the 60% AA group. This might have been due to the highest abundance of LAB in the 60% AA group, which had the potential to effectively decrease the mycotoxin contamination of silage [51]. Furthermore, AA contains large amounts of flavonoids, essential oils, and other bioactive compounds, with antibacterial, antifungal, antioxidant, and other beneficial pharmacological effects [52,53]. In line the previous microbial population results, in which the fungi in the 60% AA group were significantly inhibited, the structural and quantitative changes of fungal communities were closely linked to mycotoxins, further illustrating the inhibitory effect of AA on fungi and mycotoxins in silage. These results provide a further basis for the mechanism via which AA and its bioactive components reduce mycotoxin content.

### 2.4. Microbial Community and Correlation Analysis

#### 2.4.1. Microbial Species Diversity

Alpha diversity, commonly measured in terms of Chao1 richness and Shannon diversity, reflects the microbial abundance and species diversity of a single sample [54]. As shown in Figure 2, the alpha diversity was higher in AA than in WS before ensiling. The reason for this may be that AA contained a higher moisture (65.81%) content and was rich in nutrients more suitable for microorganism growth.

In terms of bacteria, after 120 days of fermentation, there was no significant difference in the Shannon index among groups, except for the significantly lower index in the 60% AA group. With prolonged aerobic exposure time, the Shannon indices in all groups exhibited no significant change, except in the AA group. Furthermore, the Chao1 index in the 60% AA group decreased, indicating reduced microbial abundance. In terms of fungi, the Chao1 indices in fresh materials were higher compared to all treatment groups after 120 days of fermentation. After aerobic exposure for 12 days, the Shannon index in WS increased. Additionally, the 60% AA group had numerically lower diversity and higher richness compared to the control, potentially due to the bacteriostatic activity of AA reducing the diversity of microorganisms. Moreover, the complex microbial communities were gradually replaced by LAB. This is also in line with research showing that the relative abundance of dominant bacteria is inversely proportional to the diversity of the microbial community [55]. During aerobic exposure, the alpha diversity increased, especially in WS. However, the diversity and abundance of bacteria and fungi in AA were always lowest due to its strong antioxidant activity, especially in the 60% AA group. Following the shift from anaerobic to aerobic conditions, the improved growth of aerobic microorganisms led to a noticeable increase in microbial diversity and richness. The lower diversity in the 60% AA group was likely due to the relatively lower pH values in inoculated silages, exhibiting antagonistic activity toward other bacteria. This change promoted a reduction in bacterial and fungi diversities, ultimately improving feed quality [32].

The difference in microbial communities in the silage was elucidated on the basis of beta diversity. The bacterial communities of the various groups can clearly be distinguished in Figure 3, which indicates that the raw materials had an impact on the microbial community composition. The microbial populations in all groups were distinct from those in the fresh materials at the end of the fermentation process. The bacterial community of CL was overall similar with that of the 60% AA group after 4 days of aerobic exposure (Figure 3a–c), with the difference becoming increasingly visible with the extension of aerobic exposure time. In terms of the fungal communities (Figure 3d–f), the compositions in the WS, CL, and 60% AA groups were overall similar after aerobic exposure for 4 days. The differences among groups gradually became noticeable, except for the CL and 60% AA groups, which converged, indicating similar mechanisms of action. These results indicate that the addition of 60% AA affected the bacterial and fungi community, boosting the fermentation quality when the environment changed from anaerobic to aerobic, with the fluctuations in microbial community potentially explaining the difference in silage quality [19].

#### 2.4.2. Bacterial Community and Correlation Analysis

Differences in microbial communities may be a key factor influencing the variation in silage quality [56]. In terms of the bacterial community (Figure 4a), the predominant bacteria of all groups shifted significantly from Proteobacteria to Firmicutes after 120 days of fermentation. Firmicutes is the dominant phylum in the fermentation of silage. It is an important acid-hydrolytic microorganism in anaerobic environments [57,58], probably because anaerobic and acidic environments benefit the growth of Firmicutes during fermentation [59]. Meanwhile, the 60% AA group had a richer Firmicutes community than other groups from the beginning (72.56%) to the end of aerobic exposure (86.44%). Thus, this result consolidated the dominant position of *Lactobacillus*, showing the boosting effect of LAB on the structure of the microbial community in the 60% AA group. These findings are consistent with the previous results, explaining the better quality of silage in the 60% AA group.

Bacteria such as *Lactobacillus*, *Enterobacter*, and *Weissella* are the main microbial species involved in acid-producing fermentation during storage, which have different tolerances to acid stress [60]. In higher-quality silage, *Lactobacillus* typically dominates, converting plant carbohydrates into organic acid; this lowers the pH of silage after plant cells and aerobic microorganisms consume oxygen in the early stage of fermentation, inhibiting the growth of *Enterobacteria* and other undesirable microorganisms such as yeast and mold [61]. Additionally, strains belonging to *Lactobacillus* are always used as silage additives to improve the fermentation quality [62]. On a genus level (Figure 4b), *Enterobacter* and *Pantoea* were the main microorganisms in the raw materials. After 120 days of fermentation, *Sporolactobacillus* and *Lactobacillus* abundances increased, whereas the *Pantoea* population decreased in all four analyzed groups. Among them, the WS group had the highest abundance of *Sporobacteria* (28.75%), while the 60% AA group had a higher abundance of *Lactobacillus* (22.39%) and *Pantoea* (0.95%), genera widely present in feed crop and silage, as also observed by Wang et al. [19]. *Pantoea* was negatively correlated with NH_3_-N (*r* = −0.26, *p* < 0.05), AFB1 (*r* = −0.51, *p* < 0.001), and DON (*r* = −0.53, *p* < 0.001) (Figure 4c), consistent with a previous study on mycotoxin content [55]. This finding may have been the reason for silage in the 60% AA group having relatively lower pH, as well as mycotoxin and NH_3_-N contents.

During aerobic exposure, the genera *Enterobacter* and *Enterococcus* were significantly enriched in the WS group, whereas their levels were lowest in the 60% AA group. Generally, *Enterobacter* is undesirable in silage because it competes with LAB, attenuating the decline in pH and accumulation of NH_3_-N [63]. In addition, *Enterobacter* was present at an exceptionally low relative abundance in the 60% AA group. This was most likely due to its low acid tolerance or the bacteriostatic impact of AA. Although *Enterococcus* represents a normal constituent of the intestinal flora of humans and animals, some studies in recent years have revealed that it has obvious pathogenicity [64,65,66,67]. In this study, *Enterococcus* was positively correlated with AFB1 (*r* = 0.63, *p* < 0.001) and DON (*r* = 0.60, *p* < 0.001). Fewer *Enterobacter* and *Enterococcus* were observed in the 60% AA group, consistent with the lowest mycotoxin content. After 12 days of aerobic exposure, *Lactobacillus* abundance markedly decreased in WS (1.89%) and AA (0.06%), whereas it remained the highest in the 60% AA group (16.75%). *Lactobacillus* was positively correlated with lactic acid level (*r* = 0.34, *p* < 0.01), but negatively correlated with pH (*r* = −0.79, *p* < 0.001), in agreement with previous results.

#### 2.4.3. Fungal Community and Correlation Analysis

In the fresh materials, the dominant phyla of WS and AA were Ascomycota and Basidiomycota, respectively (Figure 5a). After 120 days of fermentation, the abundance of Ascomycota increased, with the lowest level of 73.53% in WS and the highest level of 93.91% in AA. With the prolongation of aerobic exposure, Ascomycota took the dominant position among all groups after 4 and 12 days, as described by Wang et al. [68]. The abundances of Ascomycota and Basidiomycota did not differ significantly between the control and inoculated silages, indicating that the inoculant had no marked effect on the composition of fungi at the phylum level during fermentation. At the genus level (Figure 5b), *Sporidiobolus* (40.91%) occupied a dominant position in raw WS, while the fungal community in raw AA was relatively rich, potentially related to its rich nutrient content. After 120 days of fermentation, the abundance of *Issatchenkia* and *Aspergillus* increased. Compared with other groups, 60% AA had the highest *Issatchenkia* abundance (69.17%) and lowest *Aspergillus* abundance (6.18%), in contrast to WS. *Issatchenkia* was previously found to increase the total phenolic level, flavonoid contents, and enhanced antioxidant activities [69]. Additionally, *Issatchenkia* was negatively correlated with pH (*r* = −0.88, *p* < 0.001) but positively correlated with lactic acid content (*r* = 0.38, *p* < 0.05) in the present study (Figure 5c). This may have been the reason for the lower *Aspergillus* abundance and mycotoxin contents in 60% AA. With the extension of the aerobic exposure time, the richness of *Wickerhamomyces* and *Aspergillus* increased. After aerobic exposure for 12 days, the abundance of *Aspergillus* was higher in WS (15.94%) and CL (19.15%), whereas it maintained a lower level in 60% AA group. Meanwhile, the 60% AA group had the highest *Wickerhamomyces* abundance. *Aspergillus* species are significant plant pathogens with the ability to produce wide varieties of secondary metabolites and induce opportunistic mycoses [70]. *Aspergillus* has been isolated from multiple materials worldwide, and the prevalence of this species in silage is highly variable, ranging from 8% to 75% of samples [71]. This finding is in line with the positive association between *Aspergillus* and AFB1 content (*r* = 0.36, *p* < 0.05). Several studies have proven that *Wickerhamomyces* can improve the gut microbiota, enrich the relative abundance of *Lactobacillus*, and enhance the aroma richness and complexity of volatile compounds, inhibiting the production of mycotoxins [72,73]. There was no significant link between AFB1 and *Wickerhamomyces*, as supported by other studies [74,75].

## 3. Conclusions

Mixed treatments with *Artemisia argyi* could effectively improve the fermentation characteristics and structure of microbial communities on wheat straw silage, especially at a proportion of 60% AA. During aerobic exposure, WS silage mixed with 60% AA enhanced lactic acid production, increased the pH value, and reduced the NH_3_-N content. The richness and diversity of unwanted bacterial and fungal species including *Enterobacter* and *Aspergillus* were reduced. In addition, the levels of mycotoxins including AFB1 and DON decreased significantly. Overall, 60% AA has great potential for improving the quality of wheat straw silage. These results provide a reference for fully utilizing straw and Chinese herb resources, reducing competition between livestock and humans for food.

## 4. Materials and Methods

### 4.1. Raw Material Collection and Production

Qiule 18 (cultivated variety) WS and whole crop Tangyin Beiai (cultivated variety) AA were harvested in Zhengzhou and Anyang, Henan Province, China, respectively, and cut to approximately 2–3 cm. As inoculants, *Lactobacillus* (*L.*) *buchneri* and cellulase were obtained from China Agricultural University and Beijing Hongrun Baoshun Technology Co., Ltd., respectively. Single colonies of *L*. *buchneri* were grown in de Man, Rogosa, and Sharpe (MRS) medium for 12 h at 37 °C, after which the culture was centrifuged at 12,000× *g* for 10 min at 4 °C. Next, the precipitate was mixed with distilled water until the OD_600_ was 0.80, while cellulase was also dissolved in sterile water.

Experimental groups were designed as follows: (1) WS (WS with no additive); (2) CE (WS with 1 g/kg cellulase); (3) LB (WS with 2% *L. buchneri*); (4) CL (WS with 1 g/kg cellulase and 2% *L. buchneri*); (5) 20% AA (WS with 20% AA); (6) 40% AA (WS with 40% AA); (7) 60% AA(WS with 60% AA); (8) 80% AA (WS with 80% AA); (9) AA only. Water content was adjusted to 65%, and experiments were set for small-scale silage of 0.5 kg per bag for 120 days of fermentation, followed by 2, 4, and 12 days of aerobic exposure at room temperature (20–36 °C). Pre- and post-fermentation samples were taken for analysis.

### 4.2. Fermentation Characteristic, Microbial Population, and Mycotoxin Content Analysis

After 120 days of fermentation, 10 g of each silage sample (FW basis) was diluted with 90 mL of distilled water, and then filtered through four layers of medical gauze. The pH value of the filtrate was determined immediately using a pH meter (Mettler Toledo Co., Ltd., Greifensee, Switzerland). According to Wang et al. [19], the concentrations of organic acids and NH_3_-N were determined using high-performance liquid chromatography (HPLC, Waters Alliance e2695, Waters, MA, USA) and the phenol/hypochlorite colorimetric method, respectively. The analytical conditions of the HPLC were as follows: column, Carbomix H-NP10 (Sepax Technologies, Inc., Newark, DE, USA, 8%, 7.8 × 300 mm); detector, diode array detector (DAD), 214 nm (Agilent Technologies Co., MNC, Santa Clara, CA, USA, Agilent 1200 Series); eluent, 2.5 mmol/L H_2_SO_4_ (0.6 mL/min); temperature, 55 °C [19].

Another 10 g of each silage sample was oven-dried to calculate the dry matter (DM) and then ground through a 1.0 mm sieve for chemical analysis. The milled sample was used to analyze the concentrations of crude protein (CP), neutral detergent fiber (NDF), acid detergent fiber (ADF), and water-soluble carbohydrate (WSC). The CP was determined using a Kjeldahl apparatus (K9860, Hainon, Yantai, China) following the procedure of the Association of Official Analytical Chemists [76]. The contents of NDF and ADF were measured according to Van Soest et al. [77]. The WSC was analyzed on the basis of anthrone colorimetry using a spectrophotometer (UV mini-1240, Shimadzu, Tokyo, Japan) [78].

As described by Pang et al. [79], the microbial populations of the materials and silages were analyzed using the plate count method. Briefly, 10 g samples (raw materials and silage) were immediately blended with 90 mL of sterile normal saline solution and gradually diluted from 10^−1^ to 10^−5^. LAB, aerobic bacteria, Bacillus, and yeast were incubated, and colony numbers were counted using MRS, NA, EMB, and PDA, respectively. The microbial colonies were quantified as viable counts of microorganisms in colony-forming units per gram (cfu/g) of FM.

On the basis of the above results, the AA mixed treatment group with the best fermentation quality was selected for a subsequent analysis of mycotoxin content and microbial communities, in comparison with the CL group. For the mycotoxin analysis, the concentrations of mycotoxins, including aflatoxin B1 (AFB1) and deoxynivalenol (DON), were determined using enzyme-linked immunosorbent assay (ELISA) kits provided by Lianshuo Biological Technology Co., Ltd. (AMEKO, Shanghai, China).

### 4.3. Bacterial and Fungal Community Analysis

A 10 g frozen sample was dissolved in 40 mL of sterile water; then, after homogenization, it was filtered through two layers of sterile medical gauze. Next, the gauze was rinsed with 40 mL of sterile water three times to recover the residual microorganisms, which was recycled by centrifuging at 12,000× *g* for 15 min at 4 °C after the filtrate was combined. DNA extraction and polymerase chain reaction amplification were operated according to the method described in Zhou et al. [80]. The DNA samples were paired-end sequenced using an Illumina MiSeq PE300 platform (Majorbio Bio-Pharm Technology Co., Ltd., Shanghai, China). The metagenomics strategy was barcoding.

The sequencing data were analyzed on the Majorbio Bio-Pharm cloud platform (https://login.majorbio.com, accessed on 1 December 2022). The Silva database was used to analyze the bacterial and fungal community structure at the phylum and genus levels. The alpha diversity at an OTU level was determined to evaluate species complexity according to the Chao1 and Shannon indices. Furthermore, principal coordinate analysis was applied to calculate the variance of microbial community structure. Spearman’s correlation analysis was performed to investigate the link between microbial communities and fermentation products.

### 4.4. Calculation and Data Analysis

Analyses of variance (ANOVAs) were conducted using the general linear model (GLM) procedure of the Statistical Package for the Social Sciences (SPSS Version 19.0, SPSS Inc., Chicago, IL, USA) to examine the differences between samples; significance was considered at *p* < 0.05.

## Figures and Tables

**Figure 1 toxins-15-00330-f001:**
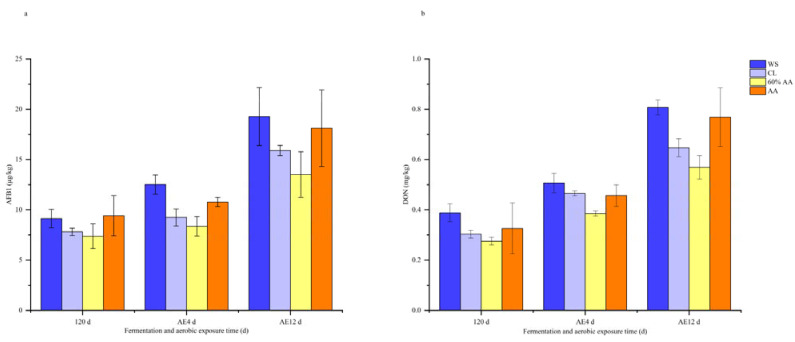
AFB1 (**a**) and DON (**b**) contents after 120 days of fermentation and during aerobic exposure.

**Figure 2 toxins-15-00330-f002:**
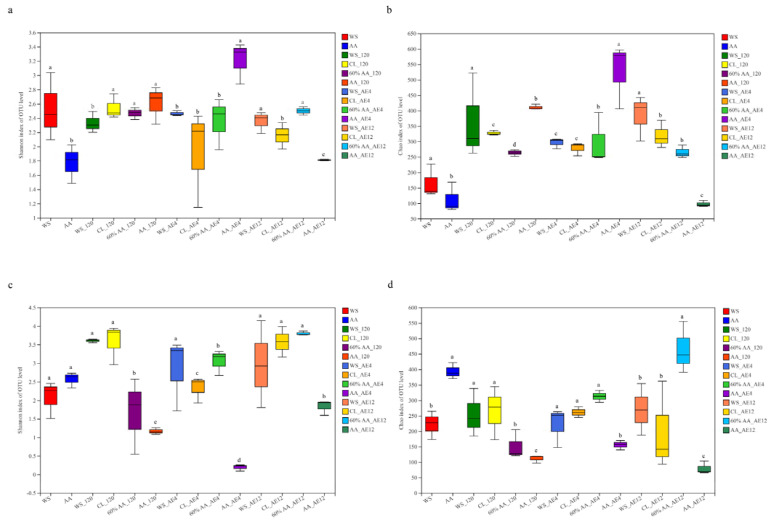
Alpha diversities at the OTU level: (**a**,**b**) Shannon and Chao1 indices of bacterial community; (**c**,**d**) Shannon and Chao1 indices of fungal community. Means with different letters in the same panel (^a–c^) have the same meaning as Table 2.

**Figure 3 toxins-15-00330-f003:**
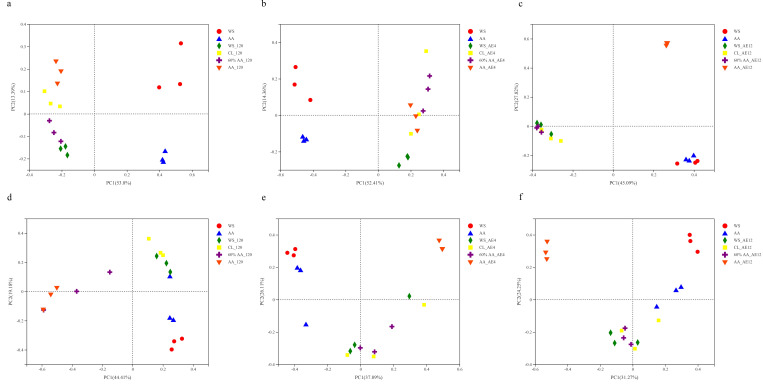
Principal coordinate analysis of microbial communities on OTU level: (**a**,**d**) bacterial and fungal communities after 120 days of fermentation; bacterial and fungal communities after (**b**,**e**) 4 and (**c**,**f**) 12 days of aerobic exposure.

**Figure 4 toxins-15-00330-f004:**
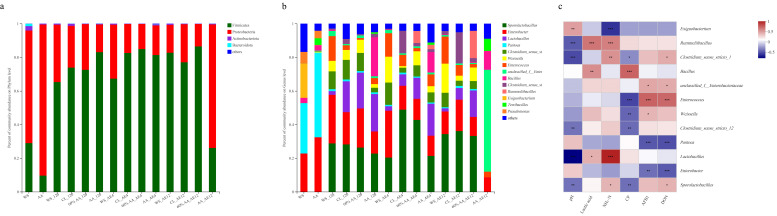
The structure of the bacterial community at the phylum (**a**) and genus levels (**b**). Correlation analysis between communities and fermentation characteristics at the genus level (**c**).

**Figure 5 toxins-15-00330-f005:**

The structure of the fungal community at the phylum (**a**) and genus levels (**b**). Correlation analysis between communities and fermentation characteristics at the genus level (**c**).

**Table 1 toxins-15-00330-t001:** Characteristics of fresh materials.

Item		WS	AA
Fermentation quality	pH	6.69	6.76
	NH_3_-N, g/kg DM	4.01	10.54
	Lactic acid, g/kg DM	1.76	2.95
	Acetic acid, g/kg DM	0.94	1.32
Chemical composition	DM, %	94.07	34.19
	WSC, g/kg DM	17.54	62.13
	NDF, g/kg DM	705.00	497.51
	ADF, g/kg DM	470.32	370.68
	CP, g/kg DM	32.24	130.64
Microbial population (log cfu/g FM)	LAB	4.43	7.13
	Bacillus	7.10	7.32
	Aerobic bacteria	7.63	9.10
	Yeast	1.45	3.36
AFB1, μg/kg		2.96	4.14
DON, mg/kg		0.08	0.11

DM, dry matter; WSC, water-soluble carbohydrate; NDF, neutral detergent fiber; ADF, acid detergent fiber; CP, crude protein; NH_3_-N, ammonia nitrogen; LAB, lactic acid bacteria; FM, fresh matter; AFB1, aflatoxin B1; DON, deoxynivalenol. No acetic, propionic, or butyric acid was detected. No mold was detected.

**Table 2 toxins-15-00330-t002:** Changes in fermentation quality during storage fermentation and aerobic exposure.

Item	Treatment	Days of Fermentation (days)					SEM	*p*-Value		
		120	AE 2	AE 4	AE 8	AE 12		T	E	T × E
pH	WS	4.91 ^aC^	5.04 ^aB^	5.14 ^aA^	5.32 ^cA^	5.00 ^cB^	0.02	<0.05	<0.05	0.08
	CE	4.92 ^aC^	5.06 ^aA^	5.00 ^aB^	5.02 ^cB^	5.11 ^cA^				
	LB	4.83 ^aA^	4.81 ^cA^	4.82 ^bA^	4.83 ^dA^	5.05 ^cA^				
	CL	4.81 ^aB^	4.77 ^cB^	4.80 ^bB^	4.78 ^dB^	5.12 ^cA^				
	20% AA	4.89 ^aB^	4.87 ^bC^	4.88 ^cC^	5.12 ^cA^	5.12 ^cA^				
	40% AA	4.81 ^aC^	4.83 ^cB^	4.83 ^cB^	4.92 ^cA^	5.08 ^cA^				
	60% AA	4.57 ^aC^	4.88 ^bA^	4.78 ^cB^	4.77 ^dB^	4.85 ^dA^				
	80% AA	4.09 ^bC^	4.32 ^dB^	4.62 ^cB^	7.00 ^aA^	7.15 ^aA^				
	AA	3.97 ^bB^	3.94 ^eB^	3.98 ^dB^	6.27 ^bA^	5.58 ^bA^				
NH_3_-N (g/kg DM)	WS	10.78 ^dA^	11.34 ^dA^	10.87 ^cA^	12.17 ^cA^	8.80 ^cB^	0.01	<0.05	<0.05	<0.05
	CE	10.96 ^dA^	13.47 ^dA^	12.92 ^cA^	11.52 ^cA^	10.15 ^cA^				
	LB	17.03 ^cB^	18.33 ^cA^	19.01 ^aA^	16.36 ^bB^	15.02 ^aC^				
	CL	18.27 ^bB^	18.50 ^cA^	17.12 ^bA^	16.86 ^bC^	16.51 ^aC^				
	20% AA	21.19 ^aA^	19.83 ^cC^	19.96 ^aC^	19.03 ^bC^	20.20 ^aB^				
	40% AA	26.01 ^aA^	23.92 ^bA^	22.97 ^aA^	21.65 ^aB^	21.36 ^aB^				
	60% AA	23.24 ^cA^	23.00 ^bA^	21.34 ^aA^	21.09 ^aA^	18.61 ^aB^				
	80% AA	25.51 ^aA^	24.24 ^cA^	23.13 ^bA^	22.18 ^cB^	20.54 ^cB^				
	AA	26.42 ^aA^	25.72 ^aA^	24.47 ^aA^	23.65 ^bB^	21.29 ^bB^				
Lactic acid (g/kg DM)	WS	2.70 ^bB^	2.14 ^bB^	1.95 ^bB^	2.59 ^bA^	1.03 ^aA^	0.08	0.22	0.12	0.38
	CE	2.20 ^bB^	3.05 ^bA^	3.27 ^bA^	3.29 ^aA^	1.67 ^aA^				
	LB	1.73 ^bA^	1.35 ^bA^	1.35 ^cA^	1.58 ^bA^	1.35 ^bA^				
	CL	1.35 ^bA^	1.35 ^bA^	1.35 ^cA^	1.35 ^bA^	1.47 ^bA^				
	20% AA	1.35 ^bA^	1.35 ^bA^	1.35 ^cA^	1.35 ^bA^	1.35 ^bA^				
	40% AA	2.35 ^bA^	1.35 ^bA^	1.35 ^cA^	1.35 ^bA^	1.35 ^bA^				
	60% AA	3.79 ^bA^	1.91 ^bB^	5.79 ^aA^	3.92 ^aA^	2.98 ^aA^				
	80% AA	9.15 ^aA^	6.99 ^aB^	4.02 ^aB^	1.35 ^bC^	1.66 ^bC^				
	AA	11.78 ^aA^	11.68 ^aA^	5.31 ^aB^	1.35 ^bC^	2.60 ^aC^				
Acetic acid (g/kg DM)	WS	2.17 ^bA^	2.58 ^bA^	2.25 ^bA^	2.56 ^cA^	2.90 ^bA^	0.11	0.32	0.18	0.55
	CE	2.74 ^bA^	2.73 ^bA^	2.33 ^bA^	2.70 ^cA^	2.79 ^bA^				
	LB	5.95 ^aB^	4.62 ^aC^	7.01 ^aA^	6.04 ^aB^	5.73 ^aB^				
	CL	4.67 ^aA^	4.63 ^aA^	3.56 ^bA^	2.64 ^cB^	5.66 ^aA^				
	20% AA	4.11 ^aA^	3.43 ^bB^	4.16 ^bA^	4.13 ^bA^	4.99 ^aA^				
	40% AA	3.89 ^aA^	3.23 ^bB^	4.40 ^aA^	2.67 ^cB^	3.95 ^aA^				
	60% AA	2.63 ^bA^	4.41 ^aA^	3.02 ^bA^	3.47 ^bA^	2.38 ^bB^				
	80% AA	3.97 ^aA^	3.22 ^bA^	1.50 ^bB^	1.29 ^dB^	1.29 ^bB^				
	AA	2.16 ^bA^	1.90 ^cA^	1.38 ^bA^	1.29 ^dA^	1.29 ^bA^				

AE, aerobic exposure time. No acetic, propionic, or butyric acid was detected. Means with different letters in the same row (^A–C^) or column (^a–d^) indicate a significant difference according to Duncan’s test (*p* < 0.05). SEM, standard error of means.; T, treatment; E, fermentation days; T × E, interaction between treatment and fermentation days.

**Table 3 toxins-15-00330-t003:** Changes in chemical composition during storage fermentation and aerobic exposure.

Item	Treatment	Days of Fermentation (d)					SEM	*p* Value		
		120	AE 2	AE 4	AE 8	AE 12		T	E	T × E
DM (%)	WS	48.74 ^aB^	51.21 ^aA^	49.95 ^aB^	59.45 ^bA^	72.40 ^aA^	3.45	1.41	0.81	2.44
	CE	44.14 ^aC^	48.01 ^aC^	50.24 ^aB^	60.36 ^aA^	69.71 ^aA^				
	LB	37.18 ^aB^	36.90 ^bB^	40.46 ^bB^	46.77 ^cA^	57.23 ^bA^				
	CL	33.19 ^bB^	34.73 ^bB^	38.54 ^bB^	39.15 ^dB^	53.67 ^bA^				
	20% AA	33.38 ^aB^	37.15 ^bB^	41.02 ^aA^	49.19 ^cA^	48.60 ^bA^				
	40% AA	30.45 ^bB^	30.46 ^cB^	32.83 ^bA^	36.93 ^dA^	39.48 ^cA^				
	60% AA	27.19 ^bB^	28.93 ^cB^	31.74 ^bB^	32.51 ^dB^	44.65 ^cA^				
	80% AA	26.56 ^bB^	26.32 ^cB^	27.27 ^cB^	32.20 ^dB^	40.53 ^cA^				
	AA	24.62 ^bB^	25.14 ^cB^	25.85 ^cB^	31.07 ^dA^	36.52 ^cA^				
WSC (g/kg DM)	WS	6.29 ^dA^	5.86 ^cA^	5.03 ^cA^	3.15 ^cB^	2.18 ^cB^	0.11	< 0.05	< 0.05	0.08
	CE	6.65 ^dA^	6.47 ^cA^	5.19 ^cA^	4.82 ^cB^	3.17 ^cB^				
	LB	4.88 ^eA^	5.62 ^cA^	4.70 ^dA^	4.12 ^cA^	2.76 ^cB^				
	CL	7.11 ^dA^	5.80 ^cA^	5.44 ^cA^	4.56 ^cB^	2.97 ^cB^				
	20% AA	6.45 ^dA^	5.62 ^cA^	5.06 ^cA^	4.41 ^cA^	3.96 ^cA^				
	40% AA	12.78 ^dA^	6.96 ^cA^	7.70 ^cA^	6.66 ^cA^	5.98 ^cA^				
	60% AA	22.34 ^cA^	11.71 ^bA^	12.90 ^cA^	10.36 ^bB^	9.62 ^bB^				
	80% AA	30.22 ^bA^	19.96 ^bB^	17.75 ^bB^	14.75 ^bC^	11.56 ^bC^				
	AA	43.22 ^aA^	40.97 ^aA^	35.67 ^aB^	33.98 ^aB^	21.72 ^aC^				
NDF (g/kg DM)	WS	664.27 ^aA^	662.37 ^aA^	660.67 ^aA^	657.98 ^aA^	656.96 ^aA^	4.36	1.77	1.02	3.08
	CE	638.27 ^bA^	635.66 ^bA^	630.38 ^bA^	626.30 ^bB^	624.80 ^bB^				
	LB	661.87 ^aA^	660.95 ^aA^	659.22 ^aB^	657.17 ^aB^	655.37 ^aB^				
	CL	648.16 ^bA^	646.58 ^bA^	645.28 ^bA^	636.94 ^aB^	633.42 ^bB^				
	20% AA	628.00 ^cA^	623.19 ^bA^	622.42 ^cA^	621.86 ^bA^	619.07 ^bA^				
	40% AA	585.24 ^dA^	582.38 ^cA^	580.20 ^dA^	577.68 ^cA^	576.34 ^cA^				
	60% AA	532.53 ^eA^	530.11 ^cA^	526.39 ^eB^	524.93 ^cB^	521.22 ^dB^				
	80% AA	516.25 ^eA^	513.05 ^cA^	512.50 ^eA^	511.67 ^cA^	509.70 ^dA^				
	AA	453.08 ^fA^	449.18 ^dA^	448.40 ^fA^	446.19 ^dA^	445.64 ^eA^				
ADF (g/kg DM)	WS	438.46 ^aA^	432.26 ^aA^	430.46 ^aA^	426.71 ^aB^	425.39 ^aB^	3.87	1.57	0.91	2.73
	CE	411.71 ^bA^	400.05 ^bA^	397.52 ^bB^	395.61 ^bB^	392.31 ^bB^				
	LB	434.75 ^aA^	427.87 ^bA^	423.74 ^aA^	420.05 ^aA^	418.83 ^aA^				
	CL	418.25 ^bA^	412.97 ^bA^	409.39 ^bB^	405.48 ^aB^	403.98 ^bB^				
	20% AA	434.29 ^aA^	432.04 ^aA^	431.18 ^aA^	429.44 ^aA^	427.85 ^aA^				
	40% AA	410.66 ^bA^	409.01 ^bA^	407.95 ^bA^	406.66 ^aA^	405.97 ^bA^				
	60% AA	376.36 ^cA^	373.68 ^cA^	370.86 ^cA^	368.88 ^cB^	367.82 ^cB^				
	80% AA	363.57 ^cA^	361.18 ^cA^	360.29 ^cA^	359.11 ^cA^	358.66 ^cA^				
	AA	350.25 ^dA^	347.23 ^cA^	346.26 ^dA^	345.28 ^cA^	344.49 ^dA^				
CP (g/kg DM)	WS	23.93 ^eA^	23.07 ^dA^	21.50 ^eA^	20.93 ^dB^	19.01 ^dB^	0.26	0.11	0.06	0.18
	CE	24.87 ^eA^	23.22 ^dA^	22.27 ^eB^	21.37 ^dB^	20.10 ^dB^				
	LB	20.40 ^eA^	18.87 ^eA^	21.70 ^eA^	19.73 ^dA^	18.87 ^dA^				
	CL	26.93 ^dA^	23.73 ^dA^	23.63 ^eA^	22.93 ^dA^	21.87 ^dA^				
	20% AA	31.07 ^dA^	31.11 ^dA^	30.30 ^dA^	30.03 ^cA^	29.73 ^cA^				
	40% AA	35.67 ^dA^	34.63 ^cA^	33.47 ^dA^	32.37 ^cB^	30.67 ^cB^				
	60% AA	49.03 ^cA^	48.00 ^cA^	47.87 ^cA^	47.53 ^bA^	48.07 ^bA^				
	80% AA	67.83 ^bA^	68.87 ^bA^	62.97 ^bA^	61.13 ^aB^	56.97 ^bB^				
	AA	81.00 ^aA^	76.87 ^aA^	75.70 ^aA^	74.33 ^aA^	72.53 ^aB^				

AE, aerobic exposure time. Means with different letters in the same row (^A–C^) or column (^a–f^) have the same meaning as Table 2.

**Table 4 toxins-15-00330-t004:** Statistics of microbial population data (log cfu/g FM).

Item	Treatment	Days of Fermnetation (d)					SEM	*p* Value		
		120	AE 2	AE 4	AE 8	AE 12		T	E	T × E
Lactic acid bacteria	WS	6.60 ^aA^	5.47 ^cB^	7.05 ^aA^	5.36 ^dB^	7.12 ^bA^	0.27	0.10	0.06	0.18
	CE	7.04 ^aA^	7.11 ^aA^	7.28 ^aA^	6.94 ^cB^	7.00 ^bA^				
	LB	6.61 ^aA^	7.03 ^aA^	5.50 ^cB^	6.79 ^cB^	6.91 ^bA^				
	CL	7.11 ^aA^	6.80 ^bA^	6.75 ^aA^	6.57 ^cB^	6.70 ^cA^				
	20% AA	7.19 ^aA^	7.41 ^aA^	6.57 ^bA^	5.13 ^dB^	6.87 ^bA^				
	40% AA	7.24 ^aA^	7.04 ^aA^	6.61 ^bA^	6.47 ^cA^	6.67 ^cA^				
	60% AA	7.27 ^aA^	7.56 ^aA^	7.30 ^aA^	9.16 ^aA^	7.64 ^bA^				
	80% AA	7.16 ^aB^	6.92 ^bB^	7.41 ^aB^	8.00 ^bA^	8.76 ^aA^				
	AA	5.19 ^bC^	6.60 ^bC^	7.50 ^aB^	7.06 ^cB^	8.97 ^aA^				
Bacillus	WS	4.50 ^bB^	5.38 ^cA^	5.90 ^cA^	5.07 ^bB^	4.94 ^bB^	0.28	0.11	0.07	0.20
	CE	5.22 ^aB^	5.39 ^cA^	6.08 ^bA^	5.27 ^bB^	5.32 ^bB^				
	LB	5.12 ^aA^	5.30 ^cA^	5.31 ^cA^	5.19 ^bA^	5.11 ^bA^				
	CL	4.80 ^aA^	4.70 ^dA^	5.65 ^cA^	4.91 ^cA^	4.55 ^bA^				
	20% AA	5.27 ^aB^	5.32 ^cA^	5.47 ^cA^	5.28 ^bB^	5.03 ^bB^				
	40% AA	5.20 ^aA^	5.34 ^cA^	5.23 ^cA^	5.55 ^bA^	6.83 ^bA^				
	60% AA	5.08 ^aB^	5.91 ^aA^	5.68 ^cB^	5.81 ^bA^	5.26 ^bA^				
	80% AA	5.18 ^aB^	6.85 ^aB^	8.65 ^aA^	9.48 ^aA^	9.49 ^aA^				
	AA	5.27 ^aC^	5.86 ^bC^	6.58 ^bB^	7.90 ^aB^	9.32 ^aA^				
Aerobic bacteria	WS	5.61 ^bC^	5.46 ^bC^	8.75 ^aA^	6.20 ^cB^	6.97 ^bB^	0.17	0.07	<0.05	0.12
	CE	5.50 ^bB^	6.71 ^aA^	6.81 ^cA^	6.77 ^cA^	7.14 ^bA^				
	LB	5.34 ^bB^	3.80 ^dC^	6.48 ^cA^	6.85 ^cA^	6.79 ^bA^				
	CL	5.35 ^bA^	4.78 ^cA^	5.38 ^cA^	5.64 ^cA^	5.31 ^cA^				
	20% AA	6.60 ^aB^	5.70 ^bB^	6.67 ^cA^	6.70 ^cA^	7.05 ^bA^				
	40% AA	5.51 ^bB^	5.49 ^bB^	6.60 ^cA^	7.20 ^bA^	6.85 ^bA^				
	60% AA	5.62 ^bB^	4.39 ^cC^	6.91 ^cA^	6.94 ^cA^	5.23 ^cB^				
	80% AA	5.37 ^bC^	7.36 ^aB^	7.59 ^bB^	9.71 ^aA^	9.62 ^aA^				
	AA	6.44 ^aC^	5.22 ^bC^	7.32 ^bB^	9.35 ^aA^	9.77 ^aA^				
Yeast	WS	2.34 ^aC^	3.70 ^aC^	4.51 ^aB^	5.23 ^aB^	8.23 ^aA^	0.08	<0.05	<0.05	<0.05
	CE	1.23 ^aB^	1.23 ^bB^	1.49 ^bB^	2.97 ^cB^	5.23 ^cA^				
	LB	1.39 ^aC^	2.09 ^bB^	1.38 ^bC^	2.39 ^cB^	3.96 ^dA^				
	CL	1.23 ^aB^	1.23 ^bB^	2.32 ^bB^	1.23 ^dB^	5.13 ^cA^				
	20% AA	ND	2.88 ^aA^	3.71 ^aA^	3.60 ^bA^	4.89 ^cA^				
	40% AA	ND	1.03 ^bB^	1.39 ^bB^	2.03 ^cB^	4.68 ^cA^				
	60% AA	ND	ND	1.23 ^cB^	1.23 ^dB^	3.86 ^dA^				
	80% AA	ND	4.58 ^aB^	4.21 ^aB^	4.21 ^bB^	6.23 ^bA^				
	AA	ND	3.42 ^aB^	3.66 ^aB^	4.36 ^bB^	7.51 ^bA^				

AE, aerobic exposure time; FM, fresh matter. Means with different letters in the same row (^A–C^) or column (^a–d^) have the same meaning as Table 2.

## Data Availability

The data generated from the study is clearly presented and discussed in the manuscript.

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
