# Peer review of "Investigation on Fermentation Characteristics and Microbial Communities of Wheat Straw Silage with Different Proportion Artemisia argyi"

_toxins, 2023, doi:10.3390/toxins15050330_

Round 1

Reviewer 1 Report

The study investigated the use of Artemisia argyi (AA) to preserve and improve the fermentation quality of wheat straw (WS) during storage, as WS is susceptible to secondary fermentation and mycotoxin poisoning.

Results for the ecological index - the authors discussed increase and decrease however figure 2 does not show any statistical difference. 

They also discussed that "Fig. 3 showed that the bacterial communities of mixture groups were clearly separated, which suggested that raw materials had an impact on the composition of the microbial community.". This is very interesting, however, the authors could better explore the differences during the fermentation of each raw material showing the increase/decrease of microorganisms and their correlation with the by-products generated during the process. 

The discussion needs to be improved to discuss how their findings could be involved in the preservation and improvement of fermentation, specifically in the addition of 60% AA which seems to be the best result since they found an increase in lactic acid production, leading to improved anaerobic fermentation. The change in the pattern of microorganisms could be deeply explored in this fermentation

Methodology section - the metagenomics strategy was barcoding or shotgun?

There are many errors in punctuation, double words, and English grammatical errors, please revise the manuscript

The figures are of low quality difficulting the analysis

Author Response

1. Results for the ecological index: the authors discussed increase and decrease however Figure 2 does not show any statistical difference.

We are sorry. Thank you for the suggestion and we have indicated the statistical difference in Figure 2 and resubmitted the figure. Please see Line 285.

2. They also discussed that “ 3 showed that the bacterial communities of mixture groups were clearly separated, which suggested that raw materials had an impact on the composition of the microbial community”. This is very interesting, however, the authors could better explore the differences during the fermentation of each raw material showing the increase/decrease of microorganisms and their correlation with the by-products generated during the process.

Thank you for the reminder. We have revised in the results and discussion.

For the raw material, “The DM content of raw WS was 94.07%, while for AA was 34.19%, which reached the ideal DM content as 30~35% for good silage [24]. Generally, 60~70 g/kg DM WSC is necessary of well-preserved fermentation, and is able to provide sufficient substrate for lactic acid fermentation [25]. The WSC in fresh AA was 62.13 g/kg DM, which was sufficient to ensure the success of the silage process [26], this is also the reason why AA fermentation quality is better. Compared with WS, AA had relatively higher contents of WSC and CP, but lower NDF and ADF, indicating that the mixed silage might provide enough nutrients.” (Line 97-105). “AFB1 levels in WS and AA were 2.96 µg/kg and 4.14 μg/kg, while DON contents were 0.08 mg/kg and 0.11 mg/kg, respectively.” (Line 111-112).

“Through 120 d of anaerobic fermentation, the pH in all groups remained below 5.00.” (Line 122), “In this study, at the end of fermentation, the content of NH3-N increased with the increase in AA applied content, probably due to the higher CP content of AA itself.” (Line 140-142), “lactic acid content changed slightly.” (Line 151-152), “ The reason why the fermentation quality of 60% AA was better could contributed to the higher number of LAB, which increased the lactic acid concentration, reduced the pH value, and thus improved the fermentation quality in the whole process of fermentation, especially in the aerobic exposure stage [35].” (Line 154-157), “In this study, an increase in the LAB produced copious lactic acid and a rapid decrease in the pH value, as well as improving the fermentation quality of 60% AA.” (Line 197-200), “For AFB1 and DON are closely related to the structure of fungal community [50], the higher number of LAB and the lower number of undesirable microorganisms in 60% AA group, may account for the lower mycotoxins contents.” (Line 234-237), “Additionally, the 60% AA group had numerically lower diversity and higher richness compared to the control, which may be due to the bacteriostatic activity of AA reducing the diversity of microorganisms.” (Line 269-271), “At the end of the fermentation process, the microbial communities in all groups were different from those in fresh materials.” (Line 292-293), “or bacterial community (Fig. 4a), the predominantly bacteria of all groups shifted significantly from Proteobacteria to Firmicutes after 120 d fermentation.” (Line 310-313), “After 120 d of fermentation, Sporolactobacillus and Lactobacillus abundances increased, while Pantoea population decreased in all four analyzed groups. Among them, WS group had the highest abundance of Sporobacteria (28.75%), and 60% AA group had higher abundance of Lactobacillus (22.39%) and Pantoea (0.95 %), a genus widely present in feed crop and silage.” (Line 332-337), “In the fresh materials, the dominant phylum of WS and AA were Ascomycota and Basidiomycota, respectively (Fig. 5a). After 120 d fermentation, the abundance of Ascomycota increased, among them WS had the lowest with 73.53% and AA the highest as 93.91%, respectively.” (Line 364-367), “As for genus (Fig. 5b), Sporidiobolus (40.91%) took a dominant position in raw WS, and the fungi community in raw AA was relatively rich, which might be related to its rich nutrients. After 120 d fermentation, Issatchenkia and Aspergillus abundance increased. Compared with other groups, 60% AA had the higher Issatchenkia abundance (69.17%) and the lowest Aspergillus abundance (6.18%); the opposite was true of WS. Issatchenkia could increase the total phenolic level, flavonoid contents, and enhanced antioxidant activities [69].” (Line 374-378).

The relationship between microbial communities and by-products of metabolic processes is described as “Additionally, Issatchenkia was negatively correlated with pH (r = -0.88, p < 0.001) and positively correlated with lactic acid content (r = 0.38, p < 0.05) in the present study (Fig. 5c).” (Line 378-381), “Numerous species of Aspergillus are important plant pathogens that can produce ranges of secondary metabolites and cause opportunistic mycoses [70]. Aspergillus has been isolated from multiple materials worldwide, and the incidence of this species in silage is highly variable, ranging from 8 to 75% of samples [71]. This was also consistent with the positive correlation between Aspergillus and AFB1 (r = 0.36, p < 0.05) content. Several studies have proved that Wickerhamomyces could improve the gut microbiota, enrich the relative abundances of Lactobacillus, and enhance volatile aroma richness and complexity, inhibiting the production of mycotoxins [72-73]. In the present study, there was no significant correlation with AFB1 and Wickerhamomyces, which is corroborated by relevant studies [74-75].” (Line 385-394).

3. The discussion needs to be improved to discuss how their findings could be involved in the preservation and improvement of fermentation, specifically in the addition of 60% AA which seems to be the best result since they found an increase in lactic acid production, leading to improved anaerobic fermentation. The change in the pattern of microorganisms could be deeply explored in this fermentation.

Thank you for the reminder. We have re-described the discussion section.

“This may have contributed to 60% AA exerting an antibacterial effect, slowing down the proliferation of aerobic bacteria and thus preventing the rapid increase in pH [30].” (Line 135-137), “ The NH3-N content in AA treated groups decreased, especially in 60% AA, until the end of aerobic exposure. This result concurred with the decrease in undesirable microorganisms such as mold and coliform bacteria.” (Line 142-145), “The reason why the fermentation quality of 60% AA was better may be due to the higher number of LAB, which increased the lactic acid concentration, reduced the pH value, and thus improved the fermentation quality in the whole process of fermentation, especially in the aerobic exposure stage.” (Line 154-156), “At the end of aerobic exposure, CP loss in 60% AA remained the lowest, which due to the lower pH inhibited the proliferation of harmful bacteria and reduced their consumption of nutrients as CP [41-42].” (Line 183-185), “In this study, an increase in the LAB produced copious lactic acid and a rapid decrease in the pH value, as well as improving the fermentation quality of 60% AA [44].” (Line 196-198) , “No yeast was detected in all AA-treated groups after 120 d fermentation in this research, while after 2 d of aerobic exposure, only 60% AA had no yeast. With the extension of aerobic exposure time, the yeast count in 60% AA remained at at a minimum level until the end of aerobic exposure. The yeast inhibition aslo might be attributed to the higher lactic acid content in 60% AA.” (Line 207-212), etc.

For the relationship between the addition of 60% AA microorganisms and fermentation quality to improve fermentation process, please refer to “Additionally, the 60% AA group had numerically lower diversity and higher richness compared to the control, which may be due to the bacteriostatic activity of AA reducing the diversity of microorganisms.” (Line 269-272), “No yeast was detected in all AA-treated groups after 120 d fermentation in this research, while after 2 d of aerobic exposure, only 60% AA had no yeast. With the extension of aerobic exposure time, the yeast count in 60% AA remained at at a minimum level until the end of aerobic exposure. The yeast inhibition aslo might be attributed to the higher lactic acid content in 60% AA.” (Line 279-283), “For fungi community (Fig. 3d-f), the compositions of fungi in WS, CL and 60% AA were becoming similar also at aerobic exposure 4 d.” (Line 296-297), “Meanwhile, 60% AA had a richer Firmicutes community than other groups from the beginning 72.56% to the end of aerobic exposure 86.44%. Thus, this result consolidated the dominant position of Lactobacillus and showing the enhancing effect of LAB on the microbial community structure in 60% AA silage. These were consistent with the previous results and explain the better quality of silage in the 60% AA group.” (Line 317-322), “Pantoea was negatively correlated with NH3-N (r = -0.26, p < 0.05), AFB1 (r = -0.51, p < 0.001) and DON (r = -0.53, p < 0.001) (Fig. 4c), which is consistent with previous study about mycotoxins contents[55]. These might be the reason that silage in 60% AA group had relatively lower pH and mycotoxin values and NH3-N contents.” (Line 337-341), “After 12 d aerobic exposure, Lactobacillus abundance markedly decreased in WS (1.89%) and AA (0.06%), while the abundance of Lactobacillus in 60% AA (16.75%) was still the highest. Lactobacillus was positively correlated with lactic acid level (r = 0.34, p < 0.01), negatively correlated with pH (r = -0.79, p < 0.001). These are all in agreement with previous results.” (Line 353-357), Line 381-382, Line 387-389, etc.

4. Methodology section: the metagenomics strategy was barcoding or shotgun?

The metagenomics strategy was barcoding, and we have marked it in the manuscript, thank you for the reminder, please see Line 458.

5. There are many errors in punctuation, double words, and English grammatical errors, please revise the manuscript.

We are very sorry. Thank you for the reminder, and to improve the quality of writing, we invited native English speaking editors to make changes to English usage, grammar, punctuation and spelling. Editorial the certificate has been submitted with the revised manuscript.

6. The figures are of low quality difficulting the analysis.

We are very sorry for the bad reading experience. We have revised all of figures in the resubmitted manuscript.

Reviewer 2 Report

Dear Authors

I believe that the manuscript deals with quite an interesting topic, but it still needs to be refined. First of all, please show in your manuscript that the research results can be applied in practice and please clearly indicate what are the conditions for this to happen.

Author Response

1. Title: Please suggest a more practical title for the manuscript. I believe that after establishing the research hypothesis it will be possible.

Thank you for the suggestion. We have modified the title for the manuscript.

Investigation on fermentation characteristics and microbial communities of wheat straw silage with different proportion Artemisia argyi

2. Line 75-79: The research hypothesis is missing from the manuscript. I will ask the Authors to propose a hypothesis. In my opinion, the hypothesis is one of the most important elements of research planning and is essential for drawing conclusions later. Please also suggest a clearer purpose of the work. In the current edition, it is hardly legible and unspecific.

We are sorry, and thank you for the reminder. We have improved this section in Line 89-94.

To sum up, mixing AA and WS together for silage may improve the fermentation quality and reduce mycotoxins production of WS silage. Therefore, the research aimed to explore the effects of different proportions AA on microbial community, fermentation characteristics and mycotoxins of WS through ensiling and during post-opening exposure process, to select optimal addition proportion AA, obtain high quality WS silage, and provide reference on solving the mycotoxins problem of fermented feed.

3. Introduction: I please enrich the Introduction with more practical aspects.

Thank you for the suggestion, and we have enriched introduction with more practical aspects, please see Line 77-86.

In recent years, AA has been increasingly used as a feed additive or fermentation raw material in animal husbandry. Wang et al. [19] added AA to whole corn straw silage, which showed that the addition of AA could reduce the levels of DON and FUM in the feed, and significantly improve the silage quality during aerobic exposure; Kim et al. [20] found that AA could regulate the gastrointestinal microorganisms and improve the growth performance and stability of meat; replacing straw with AA in sheep diet significantly improve feed intake, rumen fermentation, internal digestibility, nitrogen retention and microbial nitrogen yield, especially at medium and high AA content levels [21].

4. Results: Please rearrange the entire results description. After all, we have numerical data in the table, so please describe properly what the results really showed and try to explain why it happened. You shouldn't rewriting the numerical values. This is a mistake and an incorrect presentation of the research results. An additional facilitation is the combination of the Results section with the Discussion, so the description of the results obtained is much simpler than with a separate description of the results and a separate discussion of these results.

Thank you very much for the reminder. We have re-described the entire results section with emphasising on what the results show and strengthening the discussion section.

5. Line 91: Did you mean bigger, not taller?

Yes, what we want to express is that WSC and CP contents in AA is bigger than that in WS.

6. Line 395-400: On what basis did the buchneriand cellulase doses be determined?

Thank you. The L. buchneri and cellulase doses were determined by the previous researches as following.

Wang Z Y, Tan Z F, Wu G F, et al. Microbial community and fermentation characteristic of whole-crop wheat silage treated by lactic acid bacteria and Artemisia argyi during ensiling and aerobic exposure. Front Microbiol, 2022, 13.

Arriola K G, Oliveira A, Jiang Y, et al. Meta-analysis of effects of inoculation with Lactobacillus buchneri, with or without other bacteria, on silage fermentation, aerobic stability, and performance of dairy cows. J Dairy Sci, 2021, 104(7): 7653-7670.

Henderson A R, Mc Donald P, Anderson D. The effect of a cellulase preparation derived from Trichoderma viride on the chemical changes during the ensiling of grass, lucerne and clover. J Sci Food Agr, 2010, 33(1): 16-20

Weinberg Z G, Muck R E. New trends and opportunities in the development and use of inoculants for silage. Fems Microbiol Rev, 2010, 1: 53-68.

Sheperd A C, Kung L. An enzyme additive for corn silage: Effects on silage composition and animal performance. J Dairy Sci, 1996, 79(10): 1760-1766

7. Line 406-408: What kind of feature/size did the authors plan to evaluate?: concentration, shape, size, activity, appearance, etc. The reviewer does not see a precise explanation of such important information. Please be specific about this information.

We are sorry and thank you for the suggestion, we have modified this part, please see Line 420-427.

The concentrations of organic acids and NH3-N were measured by high performance liquid chromatography (HPLC, Waters Alliance e2695, Waters, MA, USA) and phenol- hypochlorite colorimetric method according to Wang et al. [19], respectively. The analytical conditions of the HPLC were: column, Carbomix H-NP10 (8%, 7.8 × 300 mm, Sepax Technologies, Inc., Newark, DE, USA); detector, Diode Array Detector (DAD), 214 nm (Agilent 1200 Series, Agilent Technologies Co., MNC, Santa Clara, CA, USA); eluent, 2.5 mmol/L H2SO4, 0.6 mL/min; temperature, 55 ℃ [19].

8. Line 406: What were the analytical conditions of the HPLC analysis?

We are very sorry. we have added the analytical conditions of the HPLC analysis, please see Line 423-427.

The analytical conditions of the HPLC were: column, Carbomix H-NP10 (8%, 7.8 × 300 mm, Sepax Technologies, Inc., Newark, DE, USA); detector, Diode Array Detector (DAD), 214 nm (Agilent 1200 Series, Agilent Technologies Co., MNC, Santa Clara, CA, USA); eluent, 2.5 mmol/L H2SO4, 0.6 mL/min; temperature, 55 ℃ [19].

9. Line 411-413: Similar note as for Line 406-408. Please include this throughout the manuscript as this lack of information is unacceptable. I please do not use colloquial expressions that often appear in the text. Please use analytical/chemical/science/technical terms, this is a manuscript of scientific importance.

We are very sorry, and thank you for the suggestion. We have modified this part, please see Line 429-431.

The milled sample was for crude protein (CP), water-soluble carbohydrate (WSC), neutral detergent fiber (NDF) and acid detergent fiber (ADF) concentrations analyzing.  

10. Line 459: I quote... for mycotoxin... - what do the authors mean? Please elaborate and scientifically construct your thought.

We are very sorry for the bad reading experience. What we mainly wanted to highlight is that the 60% AA had lower levels of mycotoxins including AFB1 and DON than other groups throughout the entire trial. We have reintegrated in the conclusion part, please see Line 479.

11. Line 455-461: What is the practical effect? What solution are you writing about for research?

We are very sorry, and thank you for the suggestion, we have modified this part, please see Line 472-481.

The mixed with Artemisia argyi can effectively improve the fermentation characteristics and structural of microbial communities on wheat straw silage, especially 60% Artemisia argyi proportion. During the aerobic exposure phase, WS silage mixed with 60% AA shown as lactic acid production enhanced, pH value and NH3-N content reduced, richness and diversity of unwanted bacterial and fungi species decreased, and undesirable microorganisms including Enterobacter and Aspergillus inhibited; in addition, mycotoxins including AFB1 and DON were significantly lower. Overall, 60% AA has great potential for improving the quality of wheat straw silage. These results provide references for fully utilizing straw and Chinese herb resources, and slow down the issue of competition between livestock and humans for food.

Reviewer 3 Report

The Authors submitted to Toxins an article on the altering microbial community and fermentation characteristics after ensiling and aerobic exposure of wheat straw mixed with Artemisia argyi.

The manuscript is well structured and it is written according to common scientific rules. It lends itself to fluid reading and is accompanied by an extensive and up-to-date bibliography.

I would like to suggest to the Authors to refer to the IARC classes of the mentioned mycotoxins in the introduction ( https://monographs.iarc.who.int/list-of-classifications ), and to improve the discussions with the future concrete prospects to which your interesting study opens.

Thank you for your efforts in perfecting this article!

Author Response

The manuscript is well structured and it is written according to common scientific rules. It lends itself to fluid reading and is accompanied by an extensive and up-to-date bibliography. I would like to suggest to the Authors to refer to the IARC classes of the mentioned mycotoxins in the introduction (https://monographs.iarc.who.int/list-of-classifications), and to improve the discussions with the future concrete prospects to which your interesting study opens.

Thank you for the suggestion, we have modified this part, please see Line 38-48, Line 73-77, and Line 223-229, etc.

According to the classified by the International Agency for Research on Cancer (IARC), Fusarium spp. can produce a variety of different mycotoxins such as deoxynivalenol (DON), zearalenone (ZEN), etc; among which DON is the most common mycotoxin [3]. Aflatoxin B1(AFB1), is one of the secondary metabolites of Aspergillus, belonging to Group 1 according to IARC, has carcinogenicity to humans, is commonly found in moldy grains, straw and feed, is also one of the most toxic mycotoxins. AFB1 can be discharged from livestock byproducts and accumulated in the food chain constantly, leading to carcinogenic, mutagenic and teratogenic phenomena, causing a great harm[4]. Exist of mycotoxins not only reduced the yield of wheat in a large area, seriously affecting its quality and causing major economic losses, it has also brought great trouble to the feed utilization of wheat straw (WS) [5].

Extracts and active compounds of AA show excellent antioxidant activity and bacteriostasis, which could improve the antioxidant capacity of silage, reduce pH, ammonia nitrogen (NH3-N) content and the number of undesirable microorganism, and significantly lower the mycotoxins including AFB1, DON, ZEN, ochratoxin (OA) and fumonisin (FUM) [18].

As a secondary metabolity produced by fungi, mycotoxin could remain in fermented feed even after fungi themselves have disappeared, and still make harmful on animal and human health. The transformation of mycotoxins into non-toxic metabolites using microorganisms and Chinese herbal medicine based on biodegradation enzymes or directly adding bioenzymes is currently attracting widespread interest from researchers due to the advantages of high efficiency, specificity and environmental friendliness [49].

Round 2

Reviewer 1 Report

The authors have addressed all the questions.

Reviewer 2 Report

Dear Editor

I believe that after the correction the manuscript is suitable for consideration for publication.